# Characterization of the Superplastic Magnesium Alloy AZ31 through Free-Forming Tests and Inverse Analysis

**Gillo Giuliano *** and **Wilma Polini**

Department of Civil and Mechanical Engineering, University of Cassino and Southern Lazio,
Via Gaetano Di Biasio 43, 03043 Cassino, Italy
* Correspondence: giuliano@unicas.it

**Abstract:** This work proposes a simple procedure to characterize 1.0 mm thick sheets of superplastic magnesium alloy AZ31. The simplest mathematical function that models the behavior of a superplastic material is a power law between stress and strain rate with two parameters connected to the material: K and m. First, the parameter m (variable with the strain) was defined directly by carrying out free-forming experimental tests at constant pressure and using a simple expression taken from the analytical modeling of the free-forming process. In the second step, an inverse analysis was carried out through a finite element model (FEM) and based on a numerical–experimental comparison between the results of the dimensionless height–time (H–t) curve, which made it possible to identify the variation of the parameter K in the same strain range. Once the m and K parameters were evaluated, it was possible to simulate the free-forming tests at constant pressure in the pressure range used to characterize the material. The proposed procedure to estimate m and K parameters made it possible to best match the numerical with the experimental results in terms of the dimensionless height–time curve. The difference between the forming time estimated by FEM and that measured experimentally along the H–t curve was found to be less than 9%.

**Keywords:** magnesium alloy; superplastic forming; FE model; inverse analysis

## 1. Introduction

Superplastic alloy sheet forming (SPF) is used to manufacture parts of complex shapes in one step. It is based on the phenomenon of superplasticity, i.e., the possibility of some materials reaching very high deformations if submitted to particular forming conditions. Materials with superplastic behavior have a fine-grained structure (generally 10 μm) and are stable during the forming process [1]. The grain refinement can be carried out by hot working at temperatures close to those connected with the superplastic deformation of the material (in this regard, the classes of materials subjected to this grain refinement are the following: duplex alloys such as $\alpha/\beta$ titanium alloys, duplex stainless steels, $\alpha/\beta$ copper alloys, and eutectic alloys such as Pb–Sn alloy) [2]. Some materials (such as duplex materials) can undergo grain refinement by phase separation (e.g., $\alpha/\beta$ Ti alloys and Zn–Al alloy). Other materials (such as low-alloy steels, ultrahigh-carbon steels, and Cu–Al alloys) can be subjected to grain refinement through phase transformation [2]. Techniques based on the use of severe plastic deformation are used to produce very fine grain sizes. The techniques based on this principle include equi-channel angular extrusion, high-pressure torsion, accumulative roll bonding, and friction stir processing [2]. The cyclic extrusion compression process technique was used for an aluminum-based alloy to obtain an ultrafine grain structure in [3,4].

During a superplastic-forming process, peculiar forming conditions are a process temperature higher than half of the absolute melting temperature of the material and a strain rate between $10^{-3}$ and $10^{-5}$ s$^{-1}$ depending on the material [1,2,5]. The high formability of superplastic sheets allows reducing the assemblies and the weight of the components that

are used in the transport sector (particularly in the aerospace and automotive industries) [2]. The main classes of superplastic materials used for industrial applications are aluminum and titanium-based alloys [1]. Titanium-based alloys have excellent mechanical properties (high strength/weight ratio), excellent corrosion resistance, and exceptional biocompatibility [2]. In the past, superplastic forming and diffusion bonding for titanium-based alloy systems have been used in the aerospace industry. More recently, titanium alloys have been successfully used in the biomedical sector and architectural applications [2]. Aluminum-based alloys are lightweight, have a good strength-to-weight ratio, and have outstanding corrosion resistance [2]. The primary user of aluminum-based alloy SPF has been the aircraft industry, which requires higher cost and produces fewer parts that justify the use of slower machining techniques. Some manufacturers have successfully applied superplastic forming to produce automobile parts in aluminum alloy with 50% weight reduction. Furthermore, 8090Al superplastic aluminum alloys have been employed to produce military aircraft panels, reducing weight by approximately 25% and production cost by 70% [2].

Magnesium alloy sheets cannot be formed successfully at room temperature due to a tightly packed hexagonal crystal structure. However, it has been observed that magnesium alloys, once heated, can reach high elongations even if the main deformation mechanism is not grain boundary sliding [6–10].

In recent decades, magnesium alloys have been used due to the continuous decrease in material manufacturing cost and the high strength/weight ratio, which enables structural applications. However, it has been highlighted that, at high temperatures, magnesium alloys are subjected to the phenomena of cavitation, grain growth, and interaction, which limit their deformability [11]. These phenomena are taken into account in predictive models, e.g., to predict grain growth during the forming process [12].

The disadvantage associated with the low strain rates that characterize the superplastic phenomenon (i.e., long forming times) has been overcome in recent years with the introduction of quick plastic-forming (QPF) processes. These processes (currently used for aluminum and magnesium alloys) are significantly shorter than the conventional superplastic-forming process: by increasing the strain rate and renouncing the high superplastic ductility, moderate elongations may be obtained [2,13–17]. The technological scheme of the QPF process is the same as that used for conventional superplastic forming; it requires the use of a pressurized gas, which, in place of the traditional punch used in a cold forming process, pushes the sheet inside of the die to copy its internal geometry. The sheet metal is previously fixed on the die through a blank holder and is heated until the optimal process temperature is reached and maintained [16]. Figure 1 shows a scheme of the SPF process. The figure shows a channel to put gas (generally argon) inside the forming chamber. The optimum process temperature guarantees the best superplastic properties of the material. Similarly, the pressure–time load curve must also be carefully designed to ensure optimal superplastic conditions. This operation is obtained using finite element modeling (FEM) which can provide the most accurate detailed information on the deformation process [18–22]. The optimal pressure–time loading curve was predicted, through an original algorithm, for the free-bulging process on an AZ31 magnesium alloy sheet [23]. However, in some cases, such as to determine the characteristic parameters of the material, the adoption of a constant-gas pressure cycle is not excluded during a free-forming test. Such a test requires formation through an elongated cylindrical die. In this way, the sheet assumes the shape of a bubble. It has been demonstrated that the free-forming test leads to the characteristic parameters of the material more simply and accurately than the tensile test does [24,25]. The free-forming test, compared to the latter, stresses the material in the same conditions as an industrial process does (i.e., a biaxial stress state), and their specimens are easier to produce. In contrast, it is not standardized and requires designed in-house equipment and a process control system.

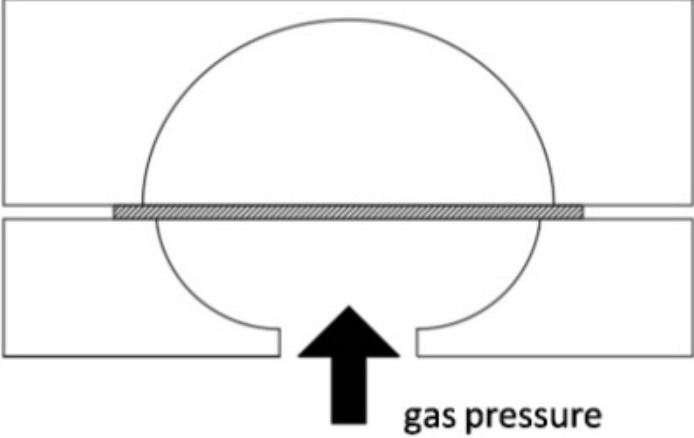

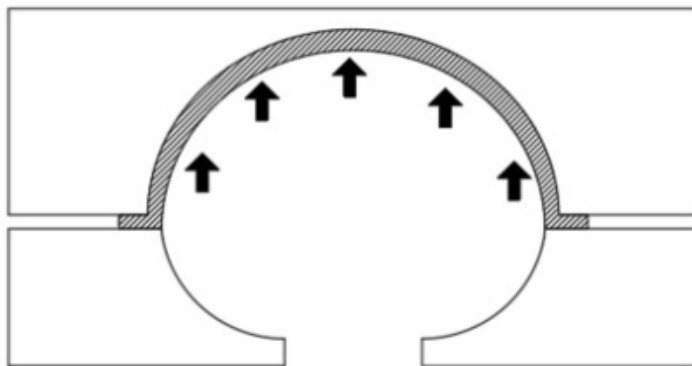

**Figure 1.** Scheme of the superplastic-forming process.

In [26], it was shown that a product with a hemispherical shape made of superplastic material, which was formed by a sheet of constant thickness, had a nonuniform final thickness distribution because of the geometric constraints. The obtained thickness distribution varied, decreasing from the edge to the pole of the sheet. In [27], it was shown that this nonuniform thickness distribution decreased as the strain rate sensitivity index m increased. Making the final thickness distribution uniform requires either the adoption of a multiphase forming process [28–31] or the use of a sheet with a variable initial thickness [16,32–35]. Multistage forming processes require designing, through finite element modeling, a preforming stage for thinning the sheet in those areas (usually the edge of the sheet) that are most stretched in the actual forming phase in correspondence with the last area of the metal sheet that comes into contact with the walls of the dies [28–31]. These preforming operations are essential even if they increase the final time of the overall formation. The use of sheets with a variable initial thickness [16,32–35] is more effective in terms of the overall forming time and weight of the component (which is partially reduced compared to the use of a sheet with a constant initial thickness) but requires a more accurate study (generally carried out using finite element analysis) to find the optimal value of the thickness distribution of the initial sheet.

It has been observed that the behavior of superplastic materials is highly sensitive to strain rate [2]. In particular, several constitutive equation models for superplasticity have been proposed, such as the uniaxial power-law model [2,36], uniaxial sinh-law model [2,37], and uniaxial superplastic-damage constitutive equations [2,38,39]. These constitutive equations contain many parameters connected with the material to characterize through experimental tests. Several mathematical functions were reported in [40]; they describe the behavior of superplastic materials for different strain rate values and different phenomena, such as work hardening and grain growth.

In the present work, a simple procedure is proposed to characterize 1.0 mm thick sheets of superplastic magnesium alloy AZ31. Through free-forming tests carried out at constant pressure and with the help of the FEM, it was possible to identify the characteristic parameters of the material present in the simplest constitutive function used in the scientific literature relating to the behavior of superplastic materials [1,2,24,41–44]. This function, which is used in a limited range of strain rates, is represented by a power law:

$$\sigma = K\delta^m, \tag{1}$$

where $\sigma$ is the flow stress, $\delta$ is the strain rate, and m and K are the material parameters. The m parameter is called the strain rate sensitivity index. This parameter has a significant role in determining the superplasticity behavior of a material. Its value is higher than 0.3; it was found to vary between 0.4 and 0.8 for superplastic materials [2]. It provides the elongation ability of a material since it gives great resistance to neck propagation during tensile deformation. According to international ISO standards, the sensitivity index to strain rate can be determined by two tensile tests carried out at different strain rates or by a tensile test at jumps of strain rate [45]. In both cases, to obtain the value of the strain rate sensitivity index, once the strain value has been preset, it is necessary to determine the stress corresponding to the impressed strain rate.

The constitutive Equation (1) cannot be used to model the grain growth and the phenomenon of cavitation. However, these phenomena depend on the equivalent strain to which the material is subjected. Therefore, Equation (1) is modified in this work by considering the parameters m and K dependent on the effective deformation, i.e.,

$$m = f(\varepsilon) \; K = g(\varepsilon). \tag{2}$$

The constitutive equation of materials represents a key factor in designing superplastic-forming processes through numerical simulation. The reliability of this equation makes it possible to determine the gas pressure profile that must be imposed on the sheet to form it inside the optimal superplastic range. The constitutive equation of the AZ31 superplastic magnesium alloy is represented by a power law that describes the relationship between stress and strain rate [46]. The parameters of the material in this law (represented by strain hardening index, strain rate sensitivity index, and strength coefficient) were considered constants and were obtained through free-forming tests and a numerical–experimental procedure which requires an inverse analysis. The searched parameters were constant throughout the range of variability of the dimensionless displacement H. On the other hand, in the present work, the values of the parameters m and K are determined through free-forming tests, while subdividing the entire dimensionless displacement measured at the apex of the dome, H, in different ranges of H, thus indirectly representing strain.

## 2. Free-Forming Test

### 2.1. Material

Magnesium alloys of industrial interest were extensively described in [47]. The superplastic magnesium alloy used in this work was AZ31; it is characterized by a weight percentage of 3% Al and 1% Zn and an average grain size of 10 μm. The sheet is used in the form of a disc with a thickness of 1.0 mm and a diameter of 80 mm. To identify the optimal superplastic conditions, it was subjected to free-forming tests at constant pressure and for different forming temperatures. The free-forming test is schematically represented in Figure 2. The figure shows two characteristic geometric parameters: the radius of the cylindrical die, a, and the height of the dome, h.

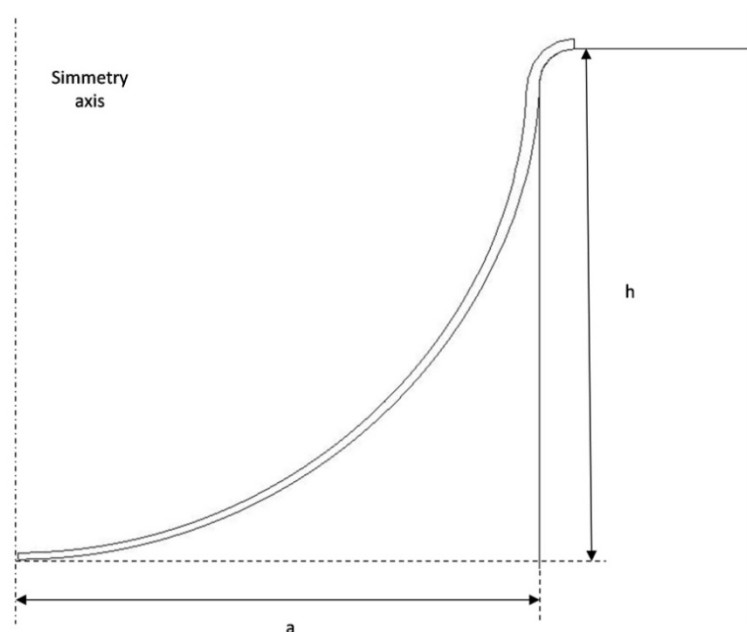

**Figure 2.** Scheme of the free-forming process (a is the radius of the cylindrical die and h is the height of the dome).

The equipment used to carry out the free-forming test is described in Section 2.2. Figure 3 shows the value of the height of the dome, once broken, for different adopted values of the test temperature and pressure. In particular, while the used test temperatures are 703, 713, 753, and 778 K, respectively, the pressure values vary between 0.2 and 0.5 MPa.

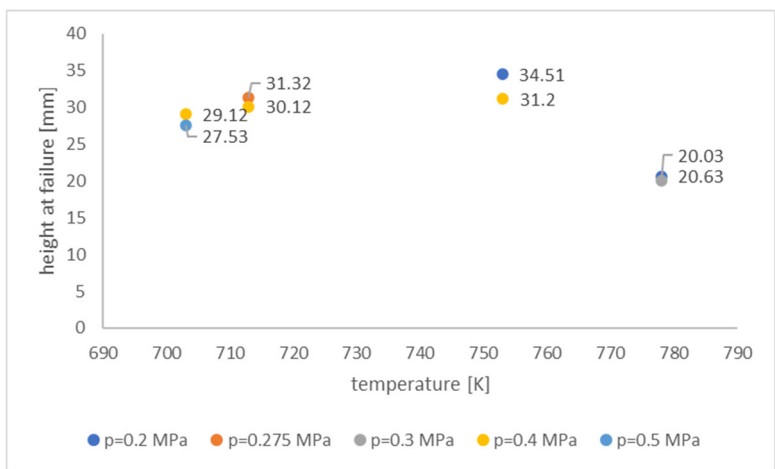

**Figure 3.** Height at failure for different temperature and pressure values in AZ31 alloy.

The adopted temperature range is suitable for the superplastic formation of the magnesium alloy under study. In each test, the pressure was kept constant, and the adopted values were chosen among those that the experimental equipment allowed applying. The figure shows that, at the temperature of 753 K the maximum height of the dome before breaking was reached. Furthermore, for each considered temperature value, increasing pressure (and, therefore, increasing strain rate) involved a reduction in the dome height at failure and, therefore, a significant reduction in the material formability. Therefore, in this work, the mechanical characterization of the material was carried out at a constant temperature of 753 K and for pressure values equal to 0.2 and 0.4 MPa.

## 2.2. Experimental Equipment

The design of the equipment used to perform the free-forming test was described in [48]. This equipment is placed on a workbench and is characterized by a die divided into two parts (Figure 4). Figure 4 presents the two half-dies and the way they are assembled.

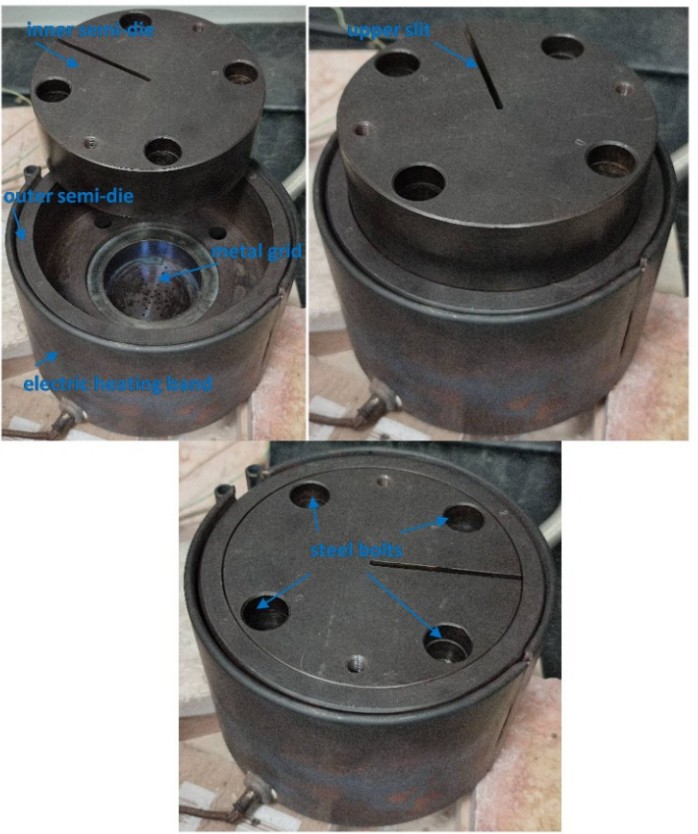

**Figure 4.** Forming die constituting two parts.

The forming sheet is located between an external cylindrical semi-die which has the task of housing the sheet; it is connected to a forming gas injection system and has a metal grid to preheat the forming gas. A second internal half-die of cylindrical shape contains the forming chamber; it rests on the sheet acting as a blank holder and has an upper slit to allow monitoring of the height at the dome apex using a measuring laser. The external half-die is wrapped in an electric heating band to indirectly bring the sheet to be formed to the forming temperature. The two half-dies are locked together using steel bolts. Some thermocouples allow monitoring of the temperature reached by the die and the sheet. Temperature control takes place on a PC using the virtual instrument Labview®. Labview® acquires the data coming from the thermocouples and acts, with the help of a control system, to maintain the temperature of the sheet to be formed. Moreover, Labview® acquires the signals coming from the measurement laser to draw the height–time curve at the apex of the dome. Figure 5 shows the bench used to carry out the free-forming test and the image of a manufactured specimen produced, starting from a circular disc. Further details on the experimental procedure are described in [48]. The forming tests were performed at constant pressure inside the range of 0.2–05 MPa. Since this work requires measuring only macroscopic parameters (dimensionless displacement curve measured at the apex of the dome vs. time, H–t) during the experimental tests, it was not necessary to carry out microstructural investigations on the specimens subjected to the forming process.

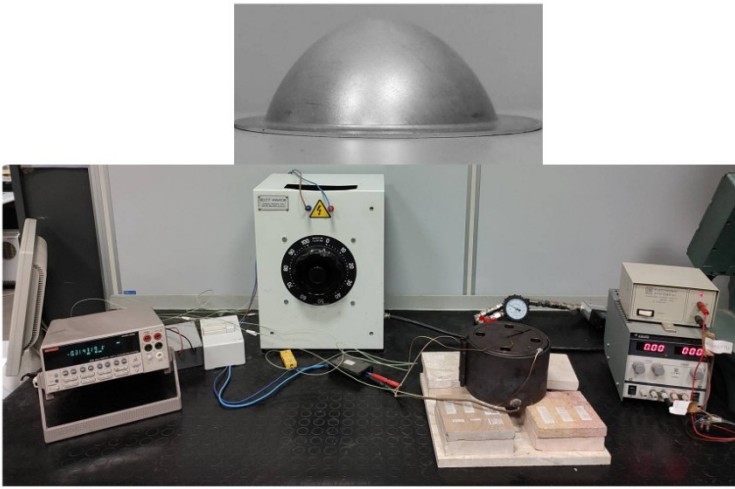

**Figure 5.** Free-forming process equipment and manufactured product.

### 2.3. Finite Element Modeling

Finite element modeling was carried out using the MSC.Marc® (Hexagon version 2005) nonlinear finite element software package and employing the rigid–plastic flow formulation. It used an axisymmetric model with full integration of isoparametric quadrangular elements. The metal sheet, which constitutes the deformable body of the problem, was divided into 128 elements. Figure 6 shows the FEM scheme of the free-forming test.

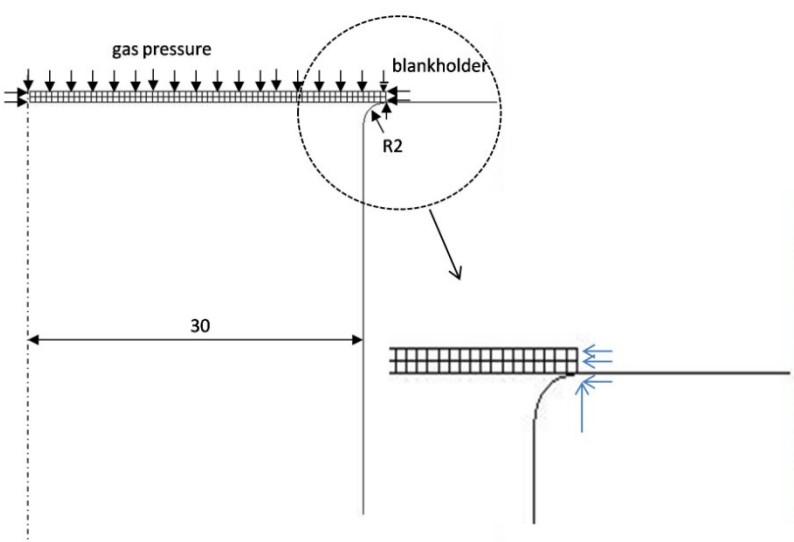

**Figure 6.** FEM representation of the free-forming process.

In the scheme, there is a die on which the metal sheet is located, which was modeled as a rigid body. To simulate the presence of a blank holder, the nodes on the outer edge of the sheet were suitably constrained. In particular, the movement of the node on the edge of the metal sheet in contact with the die was locked. The further nodes on the metal sheet edge were constrained not to move in the radial direction. Figure 6 shows, through a zoomed-in view, how the boundary conditions were applied on the outer edge of the sheet. Similarly, the nodes of the metal sheet on the axis of symmetry had a zero displacement along the direction perpendicular to the axis of symmetry. An evenly distributed pressure was applied to the upper edges of the sheet. The constitutive function of the magnesium superplastic alloy material and the forming pressure value were introduced by the use of a subroutine. The $m$ and $K$ parameters of Equation (1) were related to the equivalent strain as described in Equation (2).

### 2.4. Procedure to Mechanically Characterize the Superplastic Material

In this work, the experimental tests were carried out at the optimal temperature of 753 K. It was necessary to determine the height-forming time curves for two different pressure levels, $p_1 = 0.2$ MPa and $p_2 = 0.4$ MPa. The height of the dome, h, was dimensionless concerning the radius of the used cylindrical die, a. Therefore, the parameter H ($H = h/a$) represents the dimensionless height measured at the apex of the dome. According to the membrane theory presented in [2,24,43,49–51], an average value of m (considered constant during the free-forming process) can be calculated using the following expression:

$$m = \ln(p_1/p_2)/\ln(t_2/t_1), \tag{3}$$

where $t_1$ and $t_2$ are the forming times needed to reach H = 1 under a constant pressure equal to $p_1$ and $p_2$, respectively.

In the present work, to consider the variation of the parameter m with the equivalent strain, the dimensionless height H was divided into several intervals ($0 \leq H \leq 0.2$; $0.2 < H \leq 0.4$; $0.4 < H \leq 0.6$; $0.6 < H \leq 0.8$; $0.8 < H \leq 1.0$). In this way, the m value, for each interval of H, was determined using an equation of the type

$$m = \ln(p_1/p_2)/\ln(\Delta t_2/\Delta t_1), \tag{4}$$

where $\Delta t_1$ and $\Delta t_2$ represent the time intervals connected with each interval of H under the action of a constant pressure equal to $p_1$ and $p_2$ respectively.

Once the value of the m parameter was calculated for each interval of H, it was possible to apply an inverse analysis to determine the value of the K parameter through different numerical simulations of the free-forming process. Starting from a value of the K parameter, the time, $t_H$, required to reach the starting dome height was determined using the FEM. At this point, the function F(K) was calculated using the following expression:

$$F(K) = \left(\frac{(t_H)^{FEM} - (t_H)^{EXP}}{(t_H)^{EXP}}\right)^2_{p_1} + \left(\frac{(t_H)^{FEM} - (t_H)^{EXP}}{(t_H)^{EXP}}\right)^2_{p_2}, \tag{5}$$

where $(t_H)^{FEM}$ and $(t_H)^{EXP}$ represent the forming times needed to reach the predefined height H, using numerical simulation and experimental data, respectively, at applied pressures $p_1$ and $p_2$. In this way, at each considered interval of H, the function F(K) was minimized to determine the parameter K. In this step, the aim was to identify the value of the K parameter which minimized the difference $(tH)^{FEM} - (tH)^{EXP}$ for the two different considered pressure values. In this case, related to the estimation of the m and K parameters for the AZ31 magnesium-based alloy, the F(K) function was varied by increasing the K value. The m and K parameters were, therefore, correlated to the equivalent strain. In particular, numerical simulation was used to identify and associate an equivalent strain value to each interval of H.

## 3. Results

In Section 2.1 it was established, as a first result, that the AZ31 magnesium alloy showed the best superplastic properties (i.e., it reached the maximum displacement at the dome apex in a free-forming process) at 753 K and for pressure values between 0.2 and 0.4 MPa (see Figure 3). These test conditions were used to carry out free-forming tests at constant pressure. Figure 7 shows the experimental dimensionless displacement–time H–t curves obtained by carrying out free-forming tests at two values of pressure, $p_1 = 0.2$ MPa and $p_2 = 0.4$ MPa, which were kept constant. Figure 7 shows that, to reach a unit dimensionless displacement (and, therefore, to obtain a hemisphere) (H = 1), doubling the applied pressure involved a reduction in the forming time by about 84%.

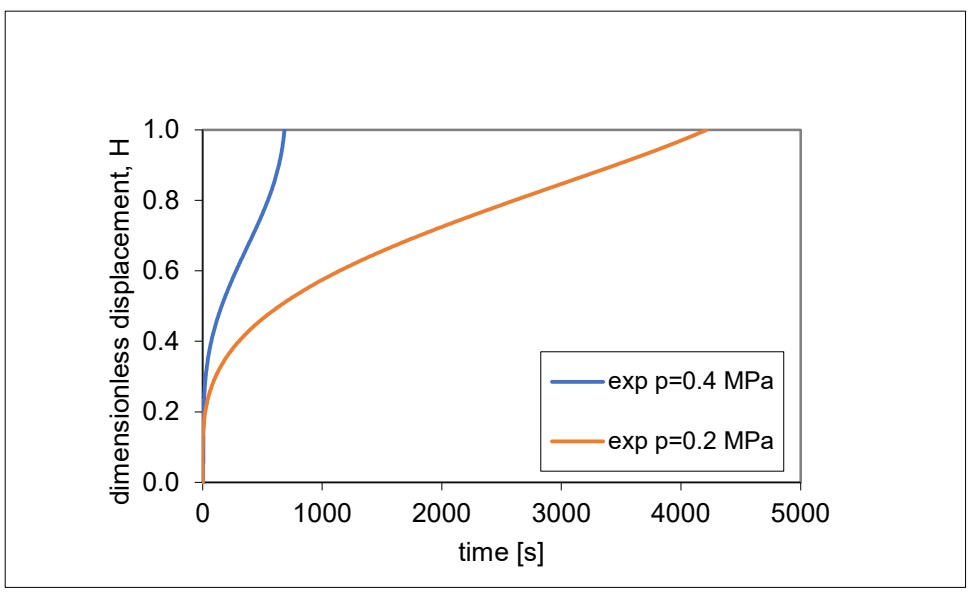

**Figure 7.** Experimental dimensionless displacement–time H–t curves.

The recording of these curves allowed determining, through Equation (4), the parameter m for different ranges of H. Figure 8 shows the obtained m–H curve.

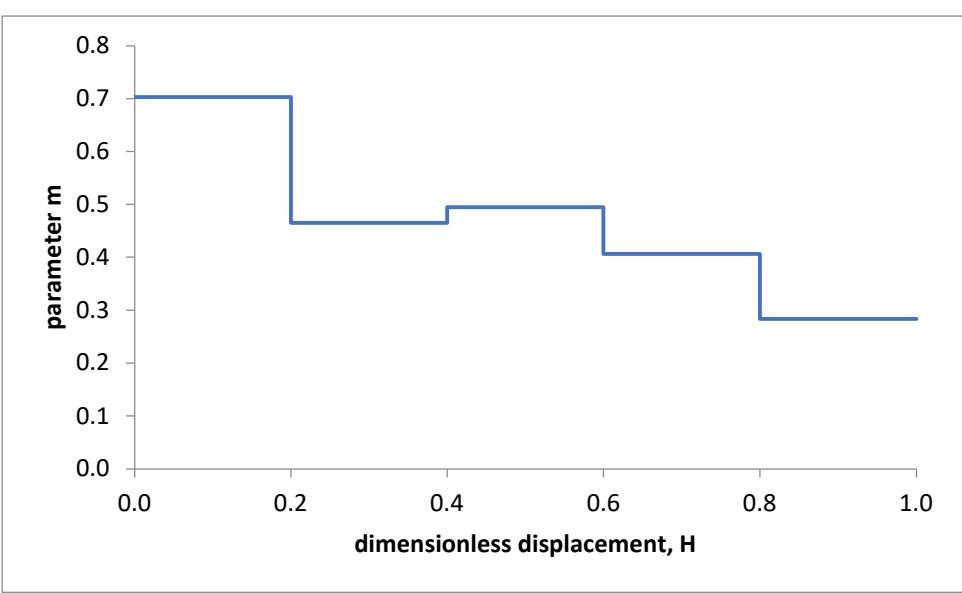

**Figure 8.** Trend of the m–H curve.

The next step required estimating the value of the K parameter. For this purpose, once the value of m was known for each interval of the displacement reached at the apex of the dome (Figure 8), an inverse analysis was applied using different numerical simulations of the free-forming process carried out for different values of the K parameter. For each interval of the reached dimensionless displacement, starting from a preset value of the K parameter, the corresponding forming time, $t_H$, was determined using FEM for each used value of K. Through Equation (5), the F(K) function was calculated for each interval of the dimensionless displacement measured at the apex of the dome. Figure 9 shows, for each adopted interval of H, the trend of F(K) as the parameter K varied.

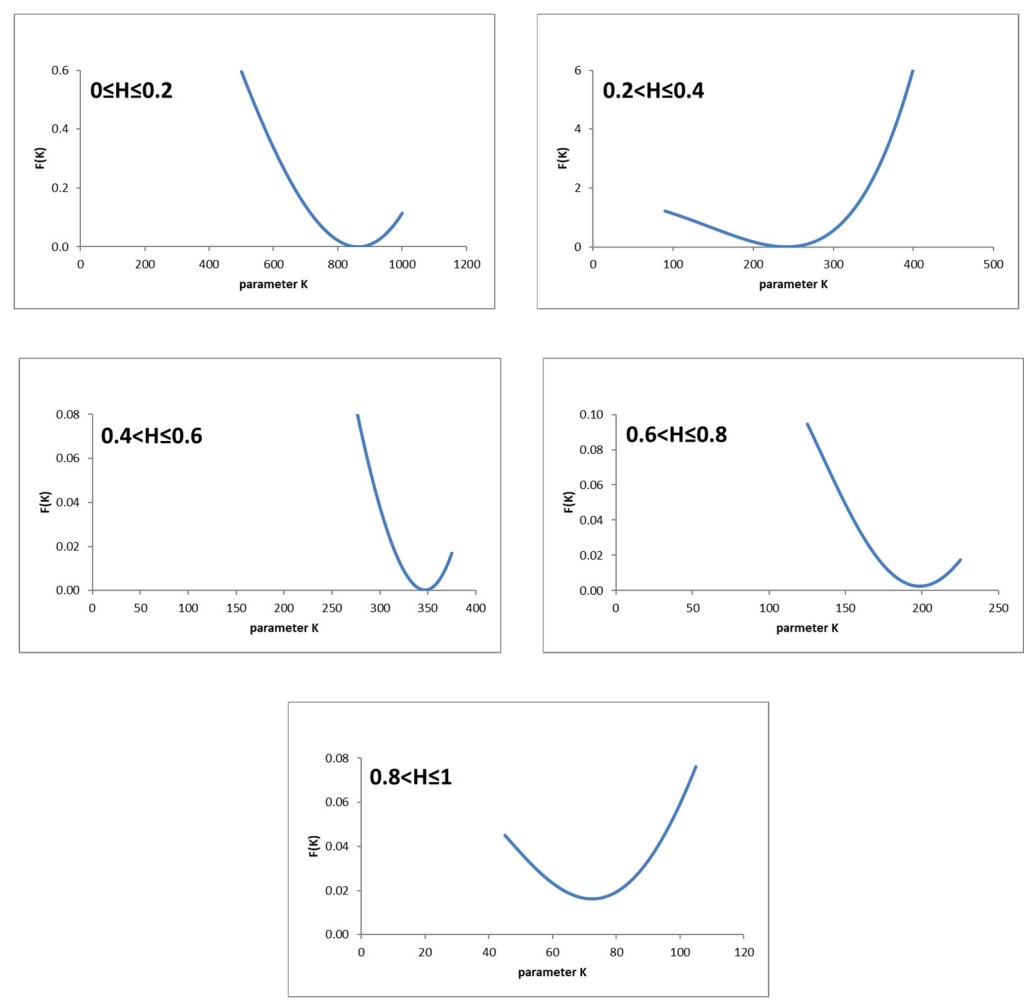

**Figure 9.** Trend of F(K) function for each adopted interval of H.

From Figure 9 it can be seen that the minimum of the F(K) function was zero when $0 \leq H \leq 0.6$, while the function assumed a value of approximately 0.0025 for $0.6 < H \leq 0.8$ and approximately 0.016 for $0.8 < H \leq 1$. Figure 10 highlights the trend of the K–H curve.

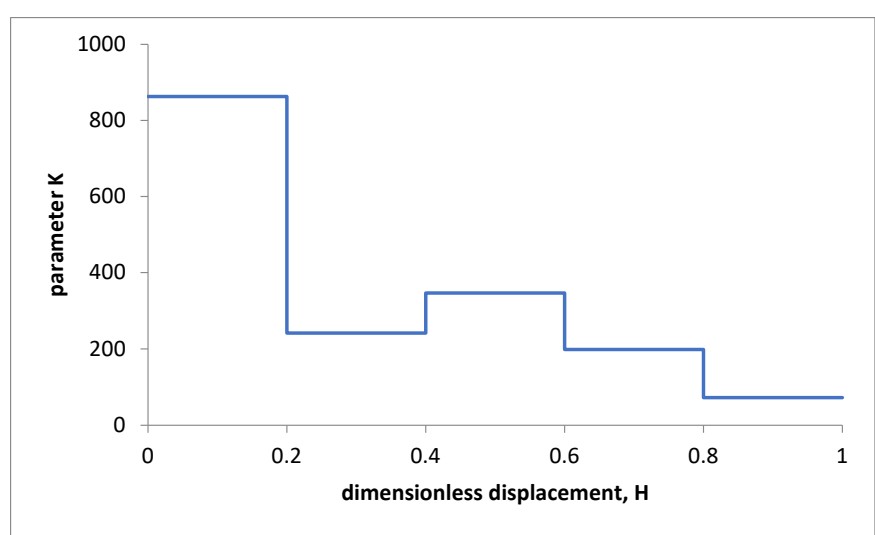

**Figure 10.** Trend of the K–H curve.

## 4. Discussions

The results of the characterization procedure of the AZ31 superplastic magnesium alloy are shown in Figures 8 and 10. The results are represented in terms of m–H (Figure 8) and K–H (Figure 10) curves. Figure 8 underlines that the parameter m tended to decrease during the free-forming process (and, therefore, with the equivalent strain) except for a slight increase for $0.4 < H \leq 0.6$. Even the parameter K tended to decrease with the dimensionless displacement, H, during the free-forming process, except for a slight increase for $0.4 < H \leq 0.6$ (see Figure 10). To evaluate the reliability of the obtained m and K values, numerical simulations of the free-forming process were performed at constant pressure $p_1 = 0.2$ MPa and $p_2 = 0.4$ MPa. Figure 11 shows the comparison between the H–t curves obtained numerically and experimentally. Figure 11 shows that the error made in predicting the forming time concerning the experimental value was slightly more than 8% for $p = 0.4$ MPa and less than 9% for $p = 0.2$ MPa in the interval $0.2 \leq H \leq 1$.

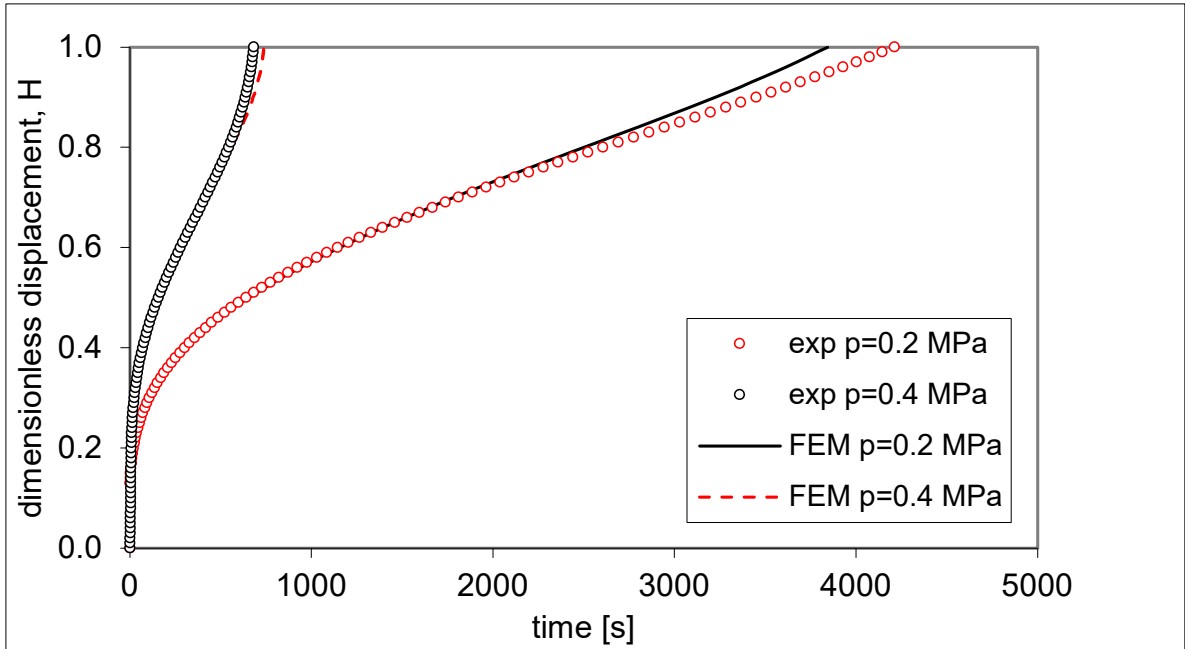

**Figure 11.** Comparison of H–t curves obtained numerically and experimentally for $p = 0.2$ and 0.4 MPa.

The constitutive function in Equation (1) with values of the parameters m and K varying according to Equation (2) was used to predict the H–t curve, even for pressure values different from those used to characterize the material. Therefore, a free-forming test with a pressure $p = 0.3$ MPa was experimentally performed and simulated by FEM. Figure 12 shows the obtained H–t curves. Figure 12 shows that the error made in predicting the H–t curve was qualitatively small. This error was just over 8% in the range of $0.2 \leq H \leq 1$. The errors evaluated in the performed tests were low (less than 9%) and assumed the greatest values in the range $0.8 < H \leq 1$, i.e., when the function F(K) did not reach zero perfectly due to the reached value of K. The maximum predicting error was determined in the range $0.8 < H \leq 1$, since the metal sheet was close to the breaking point. Near the breaking point, the behavior of the material depends on other phenomena, such as cavitation, growth of grains, and their interactions, which limit material formation.

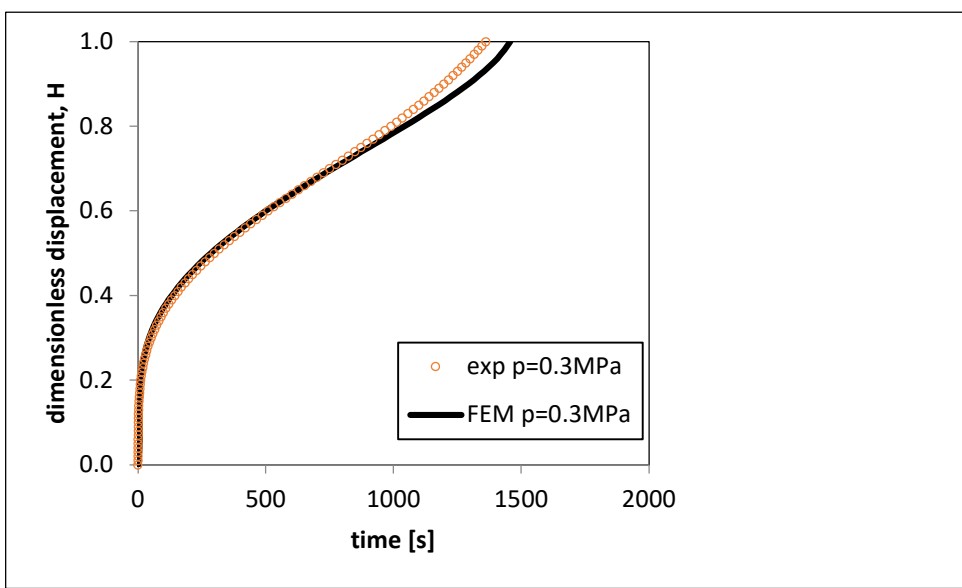

**Figure 12.** Comparison of H–t curves obtained numerically and experimentally for *p* = 0.3 MPa.

## 5. Conclusions

This work proposed a simple procedure for the characterization of 1.0 mm thick plates made of AZ31 superplastic magnesium alloy; it was based on a numerical–experimental approach. This procedure is associated with a simple equation that models the behavior of superplastic materials. The simplest mathematical equation that models the behavior of a superplastic material is a power law function between stress and strain rate. The material parameters, called m and K, were not considered as material constants (unlike the literature on the characterization of materials through the use of the free-forming test), but as parameters depending on the equivalent strain. The procedure adopted in this work to determine these parameters required a numerical–experimental approach. The experimental test consisted of a free-forming process at constant pressure whose output was represented by the H–t curve, i.e., the variation over time of the dimensionless displacement measured at the apex of the dome. Through the experimental tests, it was first possible to identify the variation of the parameter m with the dimensionless displacement. Then, using an inverse analysis based on the finite element method, it was possible to draw the variation of the K parameter with the same dimensionless displacement. Once the m and K parameters were determined, numerical simulations of the free-forming process were carried out at constant pressure, and the pressure values belonged to the range adopted for the characterization of the material. The results of the numerical simulations, in terms of the H–t curve, were compared with the experimental results of the same free-forming tests. The recorded errors were lower than 9%, thus increasing the reliability of the values of the m and K parameters determined using the simple material characterization procedure proposed in this work.

**Author Contributions:** Conceptualization, G.G. and W.P.; Methodology, G.G.; Writing—original draft, W.P. All authors have read and agreed to the published version of the manuscript.

**Funding:** This research received no external funding.

**Institutional Review Board Statement:** Not applicable.

**Informed Consent Statement:** Not applicable.

**Conflicts of Interest:** The authors declare no conflict of interest.

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
