# Peer review of "Characterization of the Superplastic Magnesium Alloy AZ31 through Free-Forming Tests and Inverse Analysis"

_applsci, doi:10.3390/app13042730_

Round 1

Reviewer 1 Report

1. The Introduction is not focused. The problem definition and novelty of the work is not clear.

2. 'm' is the strain rate sensitivity and it has significant role in determining the superplasticity of any material. But that is not clearly mentioned or discussed in the Introduction.

3. The results are not properly described or discussed.

4. As the work contains experimental part also, why not show the changes in microstructures.

5. The conclusions are not specific.

Author Response

Please see the upload file

Reviewer 2 Report

This paper dealt with a simple procedure to characterize the superplastic behavior of AZ31 Mg alloy sheets. In this regard, the most straightforward mathematical function that models the behavior of a superplastic material is a power law between stress and strain rate with two parameters of K and m related to the material. The obtained results seem suitable and can attract many readers. However, some suggestions and revisions to improve the quality of the manuscript should be applied before acceptance.

1) The manuscript should be revised based on the English language point of view.

2) The main novelty of this research should be added to the last part of the introduction.

3) In the abstract, add the most important results obtained. It seems that the abstract is not well presented.

4) Some papers should be added to the introduction and/or results such as:

1. 10.1016/j.pmatsci.2022.101016 (For grain refinement mechanisms during
deformation)   2. 10.1007/s12289-008-0203-0   3. 10.1063/1.3552429

5) Results and discussion should be separated. Also, this section should be completed. It is currently incomplete.

6) Conclusions should be revised based on the main findings of this work.

Author Response

Please see the upload file

Reviewer 3 Report

Reviewer Comment
Manuscript number: applsci-2198493

Dear Editor,

The manuscript discusses characterization of simple methods of Mg Alloy AZ31 through free forming tests and inverse analysis.The introduction section describes the required understanding of the manuscript.

The flow of writing and easy language make the understanding as easy as possible.

The overall quality of the submitted manuscript will be acceptable after consideration these suggestions:

1- The resolution of Fig. 1, 2, 3, 7, 8 and 11 needs improvement and to be modified.

2-in the section 2.1, the length of the sheet was not mentioned.

3-in the section 2.3, how the nodes of the outer edge of the edge were constrained?

4-is the difference between th FEM and th experimental is significant? The authors have not described it.

Author Response

Please see the upload file

Reviewer 4 Report

The current paper reports the characterization of the superplastic magnesium alloy AZ31 through free-forming tests and inverse analysis. The writing style is poor with not enough results and poor quality images. A number of times, the authors’ claims are not presented/supported by experimental proof. There is hardly any scientific explanation/mechanism beyond some experimental data which are not even adequate.  Based on that, I recommend major revision of the manuscript. The specific comments are as follows:

1.     The paper require a moderate revision from grammar and language point of view.

2.     The motivation/ research gap is not stated clearly.

3.     Abstract: It should be revised and contain the main essence of the work, without experimental details and also contain some results.

4.     Fig. 1  and others should be enlarged.

5.     Proof of the claims such as chemical composition, grain size etc. are required in the form of SEM/EBSD/ EDX data.

6.     The quality of the images must be improved. In current form they are hardly readable.

7.     Better to include the photograph of the equipment as well as the initial and final shape of the product.

8.     The introduction sections needs to be improved by stating the foreseen advantages of the proposed method and its justification.

9.     There is little to no scientific explanation regarding the presented results. The presentation of only data does not provide enough merit to the paper to get published.

10.  The conclusion section should be revised to avoid repetition.

Author Response

Please see the upload file

Round 2

Reviewer 1 Report

The manuscript is acceptable.

Reviewer 2 Report

1) Please prepare the manuscript according to the MDPI template.

2) I couldn't find the response letter, so, prepare it.

3) Add this paper to the introduction/result section: doi.org/10.1016/j.pmatsci.2022.101016

Reviewer 4 Report

Can be accepted in present form.
